# Salicylanilides and Their Anticancer Properties

**DOI:** 10.3390/ijms24021728

**Published:** 2023-01-15

**Authors:** Tereza Kauerová, María-Jesús Pérez-Pérez, Peter Kollar

**Affiliations:** 1Department of Pharmacology and Toxicology, Faculty of Pharmacy, Masaryk University, Palackého tř. 1946/1, 612 42 Brno, Czech Republic; 2Instituto de Quimica Medica (IQM, CSIC), c/Juan de la Cierva 3, 28006 Madrid, Spain

**Keywords:** salicylanilides, anticancer properties, mitochondrial uncoupling, TK EGFR, STAT3, drug repurposing, niclosamide

## Abstract

Salicylanilides are pharmacologically active compounds with a wide spectrum of biological effects. Halogenated salicylanilides, which have been used for decades in human and veterinary medicine as anthelmintics, have recently emerged as candidates for drug repurposing in oncology. The most prominent example of salicylanilide anthelmintic, that is intensively studied for its potential anticancer properties, is niclosamide. Nevertheless, recent studies have discovered extensive anticancer potential in a number of other salicylanilides. This potential of their anticancer action is mediated most likely by diverse mechanisms of action such as uncoupling of oxidative phosphorylation, inhibition of protein tyrosine kinase epidermal growth factor receptor, modulation of different signaling pathways as Wnt/β-catenin, mTORC1, STAT3, NF-κB and Notch signaling pathways or induction of B-Raf V600E inhibition. Here we provide a comprehensive overview of the current knowledge about the proposed mechanisms of action of anticancer activity of salicylanilides based on preclinical in vitro and in vivo studies, or structural requirements for such an activity.

## 1. Introduction

There is a persistent need to search for new, effective drugs in the field of anticancer therapy. A modern trend is the effort to develop drugs with a highly selective activity against tumor cells, with toxicity lower than the toxicity associated with conventional antitumor therapy. Development of new anticancer drugs is a very complicated process, which is typical for one of the highest attrition rates compared to other drug groups [1]. One of the attractive strategies for overcoming difficulties associated with drug development in oncology is based on drug repurposing (that could be also called drug repositioning, reprofiling or re-tasking). This strategy searches for new medical applications for already approved or investigational drugs in another indications. This approach is particularly advantageous due to the existing information on the safety profile of already registered drugs and can significantly reduce development costs and timelines [2,3]. Drugs from the salicylanilide group, resulting from the coupling of a salicylic acid and an aniline, are one of the promising candidates for repurposing in oncology. Over the past two decades, a number of studies have demonstrated the potential of antitumor activity in the group of anthelmintic drugs, among which salicylanilides were shown to induce cytotoxic or cytostatic effects in vitro or in vivo [4,5,6,7,8,9,10,11,12] and some salicylanilides even underwent clinical trials [13,14].

Salicylanilides found their use in medicine already in the 1940s [15,16]. Later, halogenated salicylanilides were used in both human and veterinary medicine as effective anthelmintic drugs such as niclosamide, rafoxanide or closantel [17]. Nevertheless, for many years, salicylanilides have been investigated for their diverse biological effects including antibacterial [18,19,20], antimycobacterial [21,22,23] and antifungal [24,25] activity. Gradually, further studies have significantly expanded the spectrum of salicylanilide effects including the primary evidence of potential anti-inflammatory action [26,27], but above all, the evidence of anticancer activity. Until now, a number of assumed mechanisms of salicylanilides’ anticancer effects have been described. For many years, salicylanilides have been known to be uncouplers of oxidative phosphorylation [28], but later studies suggested that mitochondrial uncoupling could be also responsible for their anticancer activity [4,6]. Compounds with salicylanilide scaffold were also shown to inhibit tyrosine kinase of epidermal growth factor receptor (TK EGFR), most likely by competition with ATP for binding at the catalytic domain of tyrosine kinase [29,30]. Finally, many salicylanilide anthelmintics were recently proven to inhibit diverse signaling pathways in cancer cells, thus inducing cell growth arrest, apoptosis, autophagy, etc. in diverse tumor cell types. Structures of salicylanilides that are currently being investigated for their potential anticancer activity are shown in Figure 1.

Regarding anticancer potential, the main attention has been paid to niclosamide so far; a summary of the most frequently proposed mechanisms of niclosamide’s anticancer effect, which are also discussed in that review, is shown in Figure 2. However, in recent years, there have been a number of studies published on the antitumor potential of other salicylanilides, which therefore also deserve our attention. A more detailed summary of studies describing their anticancer potential is provided in Table 1.

The aim of this review is to summarize the current knowledge about the potential antitumor effects of the group of salicylanilides to describe their expected mechanisms of anticancer activity and to conclude the structural requirements for such effects.

## 2. Structural Features of Salicylanilides Related to Their Biological Activity

Since the spectrum of biological activities described for niclosamide and similar salicylanilides is fairly wide, it is not simple to directly correlate the biological effects with a unique structural feature.

On one hand, salicylanilides have been proposed to exist in two major conformations: the “closed-ring” conformation, with an intramolecular hydrogen bond between the phenolic OH and the carbonyl O atom (OH · · · O=C) (Figure 3), and the “open-ring” conformation, where the hydrogen bond is established between the amido NH and the phenolic OH (NH · · · O); thus the H from the phenol is free. According to Suezawa et al., salicylanilides unsubstituted at positions 3 or 2′ and/or 6′ show a strong tendency to adopt the “close-ring” conformation, creating a pseudo six-membered ring [50]. Based on this, Deng et al. synthesized a series of salicylanilides meant to mimic quinazolines as EGFR inhibitors [51]. Interestingly, methylation of the phenolic OH abolished the inhibitory activity against EGFR, supporting the hypothesis that the “closed-ring” conformation was critical for biological activity.

It should be mentioned that the major conformation, observed in a crystal structure and in solution, can also differ [53]. Moreover, the “closed-ring”/”open-ring” equilibrium can be altered by the presence of electron withdrawing groups and/or when the salicylanilides are faced to an anion, as shown by Guo et al. [52]. Interestingly, the anthelmintics from the group of halogenated salicylanilides with proven anticancer activity in vitro and in vivo, such as niclosamide or closantel or salicylanilide IMD0345 [41], are functionalized with electron withdrawing groups at the aniline (Figure 1). The association of the presence of electron-withdrawing groups with cytotoxic or cytostatic activity in different cancer cell types was also observed in salicylanilide derivatives—ring-substituted hydroxynaphthanilides [54,55,56,57]. Furthermore, Tang et al. also report on such a connection between electron-withdrawing-group substitution pattern in niclosamide derivatives and their cytotoxic activity towards HL-60 cells [58]. However, it is difficult to correlate the presence of such groups with a major conformation. Instead, as already indicated originally by Williamson and Metcalf, these groups can be very relevant for the mitochondrial uncoupling of salicylanilides [28]. Indeed, the capacity of niclosamide to uncouple the mitochondria is related to its behavior as a protonophore [59]. This term refers to chemicals with dissociable protons (in this case the phenol) and lipophilic character, so that both the acid (the phenol) and the conjugate base (the phenolate) can cross lipid bilayers (Figure 4) [60]. This occurs because such molecules have extensive π-systems that can delocalize the negative charge of the conjugate base. Therefore, it is very possible that the presence of electron-withdrawing groups facilitates the delocalization of the negative charge, affecting the protonophoric behavior of salicylanilides, and this further impacts their cytostatic or cytotoxic activity.

## 3. Mitochondrial Uncoupling

For many years, salicylanilides were known as potent mitochondrial uncouplers [28,62,63,64,65]. Mitochondrial uncoupling is a process that enables proton transport from the mitochondrial intermembrane space into the mitochondrial matrix, independent from ATP synthase. Mitochondrial oxidative phosphorylation (OXPHOS) is a complex process leading to the generation of energy. Electron carriers NADH and FADH2 supply electrons to the electron transport chain (ETC). ETC is composed from consequent electron transfer reactions and thus induces electron transfer that enables proton pumps to transport protons from the mitochondrial matrix into the intermembrane space creating a proton (ΔpH) and electrical (ΔΨ) gradient. Then, protons can be transferred back into the mitochondrial matrix by ATP synthase to generate ATP [60,66,67].

Mitochondrial metabolism plays an essential role in processes that ensure the growth of cancer cells and their survival [68]. Most cancer cells were shown to exert an altered form of cellular metabolism, which is known as aerobic glycolysis, and this phenomenon is also called the Warburg effect. The metabolism of these cells is reprogrammed to preferentially perform glycolysis rather than oxidative phosphorylation, to intensify glucose uptake and to increase conversion of glucose to lactose, even in the presence of a sufficient amount of oxygen. Unlike in differentiated cells, where glycolysis leads to complete oxidation of pyruvate in the mitochondria, in that mode of cellular metabolism glucose does not undergo oxidation completely, pyruvate flux into mitochondria is decreased and a portion of glucose is shunted to anabolic pathways that are necessary for maintaining cell proliferation [4,69,70].

Mitochondrial uncouplers as salicylanilides are weakly acidic lipophilic protonophores that can transfer protons across the inner mitochondrial membrane (IMM) into the mitochondrial matrix [71,72,73]. Such uncoupling agents decrease the proton gradient and thus change the energy efficiency of mitochondria, resulting in i.e., a futile oxidation of acetyl coenzyme A (acetyl-CoA), without ATP production. Thus, these uncouplers break the link between mitochondrial oxidation and ATP generation [4,74].

Glucose metabolic pathways in cancer cells became interesting targets for potential anticancer therapeutics [75]. Alasadi et al. proposed mitochondrial uncoupling to be an approach for antagonizing aerobic glycolysis, and moreover affecting biomass production which is vital for cell growth. In their study, they have demonstrated that niclosamide ethanolamine (NEN)—a salt form of niclosamide with ameliorated bioavailability, which also acts as a mitochondrial uncoupler—can intensify pyruvate flux into mitochondria and mitochondrial oxidation. NEN was also reported to decrease lactate production and the biosynthetic pentose phosphate pathway. The authors offer a hypothesis that mitochondrial uncoupling is responsible for antiproliferative and universal anticancer activity of niclosamide, rather than affecting a specific signaling pathway in cancer cells. Thus, the inhibition of oncogenic signaling pathways might be a result of mitochondrial uncoupling [4]. Other studies also showed induced mitochondrial uncoupling by niclosamide in cancer cells [59], including the selectively stimulated generation of mitochondrial superoxide as a specific form of RONS [76].

Jiang et al. further stated the hypothesis that mitochondrial uncoupling could reverse the Warburg effect and restore differentiation in cancer cells. In that study, NEN was reported to enhance the nicotinamide adenine dinucleotide (NAD)^+^/NADH ratio, pyruvate/lactate ratio and the α-ketoglutarate (α-KG)/2- hydroxyglutarate (2-HG) ratio. Moreover, NEN induced epigenetic changes and thus induced neural differentiation in neuroblastoma cells. Above that, NEN induces upregulation of p53 and downregulation of N-Myc and β-catenin signaling in neuroblastoma cells. These results support the hypothesis that the pleiotropic effects of niclosamide are rather connected with mitochondrial uncoupling, metabolic and epigenetic changes and activation of differentiation [77].

Mitochondrial uncoupling was also proven in another salicylanilide derivatives such as oxyclozanide [4,78] and closantel [6,79]. It is also worth mentioning the uncoupling properties of nitazoxanide. Although nitazoxanide does not contain the salicylanilide moiety in its structure, as a nitrothiazolyl-salicylamide derivative, it possesses a structural similarity to niclosamide [80]. Nitazoxanide, which is a prodrug of tizoxanide (Figure 5), belongs to the group of anthelmintic drugs [81]. Senkowski et al. reported nitazoxanide to be a potent OXPHOS inhibitor and suggested that nitazoxanide could be a prospective candidate for the drug repurposing process. Moreover, authors stated a hypothesis that oxidative phosphorylation can be considered as a promising target for anticancer therapy because under low glucose conditions, cancer cells depend on oxidative phosphorylation rather than on glycolysis [6]. Recently, nitazoxanide was also shown to disrupt mitochondrial function in multicellular tumor spheroids formed from Huh-7 and HCT116 cells [82].

## 4. TK EGFR Inhibition

For many years, salicylanilides have been known to be potential epidermal growth factor receptor (EGFR) tyrosine kinase inhibitors. The receptor for epidermal growth factor EGFR (also called as ErbB1 or HER1) is, together with other closely related receptors for epidermal growth factor 2, 3, and 4 (ErbB2/HER2/Neu, ErbB3/HER3 and ErbB4/HER4), part of the receptor family for ErB/HER growth factors, which affect a number of cellular processes leading to cell proliferation and survival, their migration, differentiation, etc. However, they also play a fundamental role in the process of carcinogenesis, when the signaling pathways associated with these receptors promote cell division, angiogenesis or metastasis and, conversely, inhibit apoptosis [83].

In a number of different types of solid tumors, EGFR deregulation is observed, which leads to uncontrolled activation of the receptor and associated signaling cascades. This dysregulated EGFR and connected signaling cascades then play an important role in cancer pathogenesis [84,85,86]. Therefore, targeting EGFR and thus inhibiting its activation and further signal transition has become an attractive approach for anticancer therapy [87]. The extracellular domain is accessible to monoclonal antibodies that can inhibit ligand binding, thereby preventing receptor activation. Cetuximab or panitumumab, which target EGFR, could be named here. On the contrary, the intracellular tyrosine kinase domain is a target site for low-molecular-weight inhibitors, where they block the activation of downstream signaling pathways through binding to the ATP-binding site [88,89].

Small molecule TK EGFR inhibitors, such as gefitinib or erlotinib, are 4-anilinoquinazoline derivatives. A hypothesis was stated that salicylanilides could imitate the presence of the pyrimidine ring of the quinazolines’ structure by forming a pseudo six-membered ring [90]. During the last two decades, a number of salicylanilides as potent inhibitors of the TK EGFR were reported [29,51,91,92,93,94].

## 5. Signaling Pathways Modulated by Salicylanilides

Most studies that evaluate the effects of salicylanilides in cancer cells focus on monitoring the modulation of signaling pathways. Here, we aim to summarize the most important modulations of selected signaling pathways by diverse compounds with salicylanilide scaffold.

### 5.1. STAT3

STAT proteins, to which STAT3 belongs, are a family of transcription factors that regulate the expression of genes associated with cell cycle control, cell survival and migration, angiogenesis, or inhibition of apoptosis. Later studies dealing with the role of the STAT3 protein in the process of carcinogenesis show that it is involved in the induction and maintenance of an inflammatory pro-carcinogenic microenvironment at the beginning of malignant transformation, and further during tumor development. These insights into the role of the STAT3 signaling pathway in a number of cancer-related cellular events have made this cascade a new potential target for anticancer therapy [95,96]. STAT3 proteins are found in the cytoplasm in the form of inactive monomers. Their activation can occur through the stimulation of different types of membrane receptors, such as cytokine receptors associated with Janus kinases (JAK) or EGFR. These receptors can be activated based on the stimulation of external factors (various infectious agents, ultraviolet radiation or various chemical carcinogens) or via internal stimuli using cytokines or growth factors. However, some non-receptor tyrosine kinases, such as Src, can also activate the transcription factor STAT3. This activation then leads to the phosphorylation of STAT3 monomers on the conserved tyrosine residue (Tyr705) and through the interaction between the phosphorylated tyrosine of one STAT protein and the SH2 domain of the partner protein, a dimer is then formed, which in this form is translocated into the cell nucleus. Here, it binds to specific DNA sequences and induces the transcription of target genes [95,96,97].

It has already been mentioned that STAT3 is involved in the processes of cell proliferation and survival. STAT3 supports the transition of cells between G1 and the synthetic phase of the cell cycle by inducing or possibly suppressing the transcription of genes encoding proteins involved in the regulation of individual phases of the cell cycle. Genes whose transcription is stimulated by STAT3 include *c-Myc*, *cyclin D1*, *D2* and *D3* or *cdc2*. The proto-oncogenes *Pim-1* and *Pim-2* have also been identified as target genes for the STAT3. STAT3 negatively affects the expression of some inhibitors of cyclin-dependent kinases, such as p21^Cip1^ or p27^KIP1^. Cell survival is supported by STAT3, by inducing the expression of genes encoding antiapoptotic proteins of the Bcl-2 family (Bcl-xL, Mcl-1, Bcl-w or Bcl-2) or survivin [98,99]. STAT3 also affects the expression of a number of other genes encoding proteins that are involved, for example, in angiogenesis (VEGF) or in the processes of cell invasiveness and migration (MMP-2 and MMP-9) [95].

The STAT3 cascade contains a number of potential targets for potential anticancer treatment to prevent signal transmission. In addition to the inhibition of membrane receptors or tyrosine kinases, direct inhibition of the activity and function of the STAT3 protein is offered, i.e., inhibition of its phosphorylation, dimerization by limiting SH2 domain binding, moving from the cytoplasm to the nucleus, or binding to DNA [95].

Based on the screening of a chemical library containing 1500 clinically used drugs, niclosamide was selected as highly potent small-molecule inhibitor of the STAT3 signaling pathway. Subsequently, niclosamide was reported to dose dependently suppress the Tyr705 phosphorylation of STAT3, and to inhibit the EGF-induced nuclear translocation of STAT3 in DU145 cells with constitutively active STAT3 [5]. Niclosamide was also found to act synergistically with erlotinib through inhibition of erlotinib-stimulated STAT3 phosphorylation and to synergize with erlotinib in the growth inhibition of head and neck cancer cell lines and Tu212 xenografts [100]. Further, niclosamide was reported to retrieve the sensitivity to radiation of triple-negative breast cancer (TNBC) cell lines via suppression of STAT3 and Bcl-2 and production of ROS [101], and was also able to overcome radioresistance in human lung cancer cells and in lung cancer xenografts [102]. Interestingly, STAT3 activation was found to be associated with upregulated expression of PD-L1 e.g., in lymphoma. PD-L1 expression was reported to be regulated by a STAT3 transcription factor [103]. Luo et al. showed that niclosamide, as an inhibitor of STAT3 phosphorylation, can downregulate the expression of PD-L1, most likely through the suppression of p-STAT3 binding to the promoter of PD-L1. Thus, the authors demonstrated the potential synergistic effect of niclosamide and anticancer therapeutics targeting PD-L1 [104].

A number of other studies also demonstrate inhibition of the STAT3 signaling pathway in various types of cancer cells by niclosamide [105,106,107,108,109]. The ability to inhibit STAT3 phosphorylation in other salicylanilide derivatives was also described [110]. Salicylanilide, based on hydroxynaphthalene carboxamides, was shown to suppress phosphorylation of STAT3 at tyrosine 705 [56]. Additionally, a hybrid of cyclohexanedione TUB015 and nocodazole using a salicylanilide core structure was reported to inhibit phosphorylation and nuclear translocation of STAT3 in DU145 cells [111]. Recently, nitazoxanide was also reported to be a moderate STAT3 pathway inhibitor [112].

### 5.2. Wnt Signaling Pathway

The antitumor effects of niclosamide were discovered through several targeted high-throughput screenings of large chemical libraries. In order to find modulators of Wnt-Frizzled signaling, a library of 1200 FDA-approved drugs was screened, and thus niclosamide was identified as a potential inhibitor of the Wnt signaling pathway by internalization of the Frizzled-1 receptor [113]. In a similar way, niclosamide was found to inhibit S100A4 promoter activity. In that study, niclosamide was shown to suppress the constitutively active Wnt/CTNNB1 signaling through prevention of the formation of CTNNB1/TCF transcription complex that can activate the S100A4 promoter, thus suppressing the expression of S100A4 [114].

Wnt/β-catenin signaling is one of the highly conserved pathways, which controls essential cell processes such as cell proliferation, survival, differentiation, cell fate in the embryonic development phase and others. The canonical Wnt signaling pathway controls degradation of the transcriptional coactivator β-catenin. Its proteasomal degradation is mediated by a complex consisting of protein Axin, the tumor suppressor *adenomatous polyposis coli* gene product (APC), casein kinase 1 (CK1) and glycogen synthase kinase 3 (GSK3) [115,116]. In the absence of Wnt signaling, the amount of β-catenin is kept at low levels. On the other hand, when Wnt proteins interact with surface receptors, signaling is mediated through disheveled (Dvl) and Axin proteins and GSK3β is inhibited; therefore, the degradation of β-catenin is suppressed. The β-catenin can then relocate into the nucleus to enhance the expression of target genes. Constitutive activation of Wnt signals could result in uncontrolled cell proliferation. Thus, the Wnt signaling pathway is another prospective target for anticancer agents. Diverse mutations of Wnt signaling components and thus dysregulation of the Wnt pathway are often found in many malignancies. Oncogenic Wnt signaling is often connected to a mutation e.g., of *APC* or *CTNNB1* [117,118].

Many other studies proved that niclosamide inhibits Wnt signaling in diverse types of tumor cells [109,119,120,121,122,123,124,125,126,127,128,129]. Moreover, Qu et al. reported that nitazoxanide also could suppress Wnt/β-catenin signaling through a specific target, PAD2, which is an enzyme responsible for protein citrullination, and thus induces citrullination of β-catenin [46]. Recently, a number of studies further reported that nitazoxanide downregulated the Wnt/β-catenin signaling pathway [48,130,131].

### 5.3. NF-κB Signaling Pathway

Nuclear factor kappa B (NF-κB) represents a protein transcription factor that is engaged in many biological processes. NF-κB plays a key role in the regulation of the immune system through inducing the expression of diverse mediators of immune processes. It is also associated with the regulation of inflammation, and affects cell proliferation and apoptosis. Additionally, its deregulation appears, e.g., in leukemia, Hodgkin lymphoma, as well as in diverse types of solid tumors. The inactive form of NF-κB is stabilized in association with IκB. As a response to stimuli of growth factors or inflammatory cytokines, a canonical signaling cascade is triggered, in which IKK, as catalytic subunit IκB kinase complex, phosphorylates IκBα, allowing its ubiquitination and proteasomal degradation. This process will trigger the translocation of NF-κB, as a transcriptional factor, into the cell nucleus, where it binds to specific DNA sequences, and thus regulates transcription target genes [132,133].

Salicylanilide IMD-0354 was identified as a selective inhibitor of IκB kinase (IKK)-β, which is responsible for the activation of the NF-κB signaling cascade (Figure 6). Originally, IMD-0354 was studied as a small molecule that targets IKK-β for its anti-inflammatory and antiallergic activity in chronic asthma, or for its possible usage as topical therapy for atopic dermatitis [27,134]. Nevertheless, IKK-β was proposed as potential target of both inflammatory disease and malignancies [135]. A number of studies recently demonstrated the ability of IMD-0345 to suppress NF-κB signaling in diverse types of cancer cells. IMD-0354 prevents uncontrolled proliferation of neoplastic mast cells with constitutively activated c-kit receptors, which are responsible for NF-κB activation [40]. Furthermore, IMD-0354 was confirmed to inhibit the growth of human breast cancer cell lines MDAMB-231, HMC1-8 and MCF-7 by inducing cell cycle arrest in the G1/G0 phase [8]. IMD-0354 also inhibited NF-κB activation and induced apoptosis in chronic lymphocytic leukemia (CLL) cells samples from untreated CLL patients in vitro [41]. Furthermore, the antiproliferative effect of IMD-0354 against pancreatic cancer cells was observed in vitro and in vivo [44] and IMD-0354 suppressed the survival of adult T-cell leukemia cells [42]. Additionally, in the case of niclosamide, its ability to inhibit NF-κB signaling pathways was shown in acute myelogenous leukemia [136] and in uveal melanoma cells [126].

### 5.4. mTOR

Mammalian target of rapamycin (mTOR) is a protein kinase that can phosphorylate serine/threonine and tyrosine residues. It is involved in multiple signaling pathways that regulate a wide range of biological processes, such as cell proliferation or survival, apoptosis and autophagy, immunity reactions and cellular metabolism. It is common to find deregulated activity of mTOR in diverse types of malignancies such as breast, liver, lung and prostate tumors. In cancer cells, enhanced mTOR activity is associated with induction of signaling through growth factor receptors, changes in energetic metabolism or inhibition of autophagy. Thus such an activity could result in cancer cell proliferation [139,140].

A collection of more than 3500 pharmacologically active compounds was screened to identify new modulators of autophagy. Through such screening, niclosamide was pointed out as one of the promising candidates, and its ability to inhibit mTORC1 signaling was later confirmed [141]. Fonseca et al. further studied the mechanism of mTORC1 signaling suppression induced by niclosamide. Authors showed that niclosamide neither inhibits upstream cascade signaling, nor suppresses mTORC1 catalytic activity or mTORC1 assembly. Therefore, it was suggested that cytoplasmic acidification, induced by niclosamide, results in the inhibition of mTORC1 signaling [142]. Chen et al. showed inhibition of mTOR signaling in cervical cancer cells and proposed the hypothesis that this inhibition is modulated by increased oxidative stress [143]. Similarly, it was reported that niclosamide ethanolamine and oxyclozanide can promote mitochondrial uncoupling, which promotes downregulation of mTOR activity [4]. Another salicylanilide, rafoxanide, was found to induce suppression of PI3K/Akt/mTOR signaling pathway activity in gastric cancer cells [35]. In HCT116 spheroids, salicylanilide nitazoxanide enhanced the activity of the AMPK pathway, and thus suppressed the mTOR pathway as a downstream target of AMPK [6].

### 5.5. Other Signaling Pathways or Tumor Suppressors as Targets of Salicylanilides

Niclosamide was reported to downregulate another developmental signaling cascade, the Notch pathway, which plays an important role in many essential processes such as embryogenesis, organ development and damage repair, differentiation, and many others. Nevertheless, Notch signaling is often altered in diverse cancer types e.g., in colorectal, breast and ovarian cancers, lung adenocarcinoma or T-cell acute lymphoblastic leukemia, etc., where its activation stimulates oncogenesis [144]. It was described that niclosamide was able to downregulate Notch signaling in colon cancer [145] or in the form of niclosamide-loaded Pluronic nanoparticles, it inhibited Notch signaling in hepatocellular carcinoma [146].

B-Raf proto-oncogene (*BRAF*) gene encodes a cytoplasmic kinase that participates in mitogen-activated protein kinase signaling pathway. *BRAF* is often mutated by oncogenic mutations that lead to increased activation of the mitogen-activated protein kinase signaling pathway. A major part of these mutations occurs in codon 600 of exon 15. In many tumor types, the most frequently occurring mutation is V600E. Such a mutation is responsible for enhanced proliferation and suppressed ability to trigger apoptosis [147]. The study, aiming to reposition organohalogen drugs as potent B-Raf V600E inhibitors, identified salicylanilides rafoxanide and closantel as compounds targeting B-Raf V600E [32].

Using structure-based virtual screening with docking software, rafoxanide was identified as a potential inhibitor of CDK4/6 [34]. Cyclin-dependent kinases (CDKs) are serine/threonine proteinkinases, whose activity depends on the binding with the regulatory subunit—the appropriate type of cyclin. CDKs belong to the group of the key regulators of the cell cycle including the above-mentioned CDKs and their binding partners cyclins, cyclin-dependent kinase inhibitors (CKI), transcription factors (c-Myc, AP-1, E2F, etc.), tumor suppressor proteins retinoblastoma protein (pRb) or p53, and a number of other mediators responsible for maintaining the correct course of the cell cycle. CDK4/6 with their partners, D-type cyclins, control the progression through the G1 cell cycle phase [148,149,150]. Expression of CDK4 and CDK6 is often upregulated in many tumor types. Already approved small-molecule CDK4/6-inhibitors such as palbociclib, ribociclib or abemaciclib are used for the treatment of breast cancer. Therefore, targeting CDK4/6 is an attractive approach in the treatment of cancer [151]. Shi et al. also reported that rafoxanide lowered the expression of CDK4/6, cyclin D, retinoblastoma protein (Rb) and the phosphorylation forms of CDK4/6 and Rb in their study [34].

The oncoprotein c-Myc regulates a very wide range of cellular processes, such as cell proliferation, cell cycle progression, differentiation and apoptosis. Increased expression of c-Myc or alteration in c-Myc activity occurs in 60–70% of both solid and hematological tumors [152,153]. It has been shown that the expression level of c-Myc closely correlates with the intensity of cell proliferation, and its function is essential for the maintenance of proliferation in both normal and tumor cells. In a number of models, a shift of cells from the G0 to the G1 phase of the cell cycle was observed after the increase in c-Myc expression. On the contrary, suppressed expression or inactivation of c-Myc leads to the disruption of the further progress of the cell cycle. Among the target genes of the transcription factor c-Myc, through which it stimulates cell cycle progression, are a number of genes encoding its important regulators. c-Myc promotes the expression of positive regulators of the cell cycle, such as cyclins type D, type E, A or B1, cyclin-dependent kinases (CDK1, CDK2 and CDK4) and transcription factors of the E2F family, and conversely suppresses the activity of p15, p21 and p27 inhibitors [154,155]. It is worth mentioning that c-Myc protein is encoded by the gene which is the target of the transcription factor STAT3 [156]. Likewise, activation of the Wnt/β-catenin pathway induces an increase in the levels of c-Myc [157]. Fan-Minogue et al. developed and subsequently applied a molecular imaging sensor based high throughput-screening system with a cell-based assay to screen five libraries with 5000 already existing compounds, in order to identify active compounds with anti-c-Myc activity. Based on that, nitazoxanide was selected as a highly potent c-Myc inhibitor [47].

Another screen was performed to find compounds with selective cytotoxicity in p53-deficient tumor cells. The authors proposed that niclosamide should be considered as a promising candidate for targeting p53-deficiency in cancer cells, as niclosamide selectively suppresses the growth of p53-deficient cells [59].

## 6. Synergism with Immune Checkpoint Inhibitors

Cancer immunotherapy represents a modern approach to the treatment of many types of malignancies, which uses the ability of the immune system to prevent cancer development. Unfortunately, tumor cells have developed the ability to evade recognition and subsequent elimination by the immune system. The goal of cancer immunotherapy is enhancement of the immune system’s response to induce an anticancer effect. Immune checkpoints, including programmed cell death protein 1 (PD-1), programmed cell death 1 ligand 1 (PD-L1) and cytotoxic T-lymphocyte-associated protein 4 (CTLA-4) have proven to be suitable targets for cancer immunotherapy. Their blockage results in a boost of immunologic activity against cancer development. Currently, the leading group of immune checkpoint inhibitors are blockers of anti-PD-1/PD-L1. Nevertheless, such inhibition itself is not always sufficient to achieve the desired antitumor effect. Recently, interesting approaches that can possibly enhance the response to immunotherapy have been proposed, such as the combination of immunotherapy with STAT3 inhibitors. This assumption was based on the fact that STAT3 stimulates the expression of immune checkpoint molecules i.e., PD-L1. Therefore, it has been hypothesized that STAT3 inhibitors may act synergistically with PD-1/PD-L1 monoclonal antibodies, as it is illustrated in Figure 7 [158,159,160].

Luo et al. showed that niclosamide, as a suppressor of STAT3 phosphorylation, downregulates PD-L1 expression in non-small cell lung cancer cells. Authors further proved this downregulation related to the blockage of STAT3 phosphorylation and further binding to the PD-L1 promoter. Thus, such a combination therapy of niclosamide and PD-1/PD-L1 inhibition can represent a possible strategy to enhance immunotherapeutic efficacy [104]. Recently, downregulation of PD-L1 expression induced by niclosamide was also demonstrated in pancreatic cancer cells [162].

## 7. Cancer Clinical Trials and Pharmacokinetic Properties

A large number of promising preclinical studies with salicylanilides or with their structurally related compounds focusing on the anticancer activity resulted in multiple clinical trials. Their list is given in Table 2. In most cases, it is niclosamide that is evaluated in these studies. However, there was already a clinical trial in which nitazoxanide was included. In this trial, NCT02366884, combinations of selected antibacterial, antifungal and antiprotozoal agents, including nitazoxanide, conventionally used for the treatment of bacterial, fungal or protozoal infections were used to treat patients with advanced, metastatic and terminal cancers.

The series of clinical trials with niclosamide encompass both trials of phase 1 and trials of phase 2. The aim of trial NCT02687009 was to define the maximum tolerated dose of niclosamide and to obtain the information about alterations in WNT signaling induced by niclosamide in humans. In addition, it is worth to mention the trial NCT02532114, as it was a dose-escalation study evaluating oral administration of niclosamide with enzalutamide in men with castration-resistant prostate carcinoma (CRPC). The results showed poor oral bioavailability of niclosamide. Unfortunately, even at its maximum tolerated dose of 500 mg, its achieved plasma levels did not occur steadily above the required threshold reported to inhibit the growth of the CRPC model [13]. Low oral bioavailability is one of the major problems in the repurposing process with niclosamide. However, a number of attempts to enhance the pharmacokinetic properties of niclosamide already exist. For example, to ameliorate its bioavailability, niclosamide-loaded submicron lipid emulsions were developed [163]. In another study, nanoliposomal encapsulation was reported to increase the aqueous solubility of niclosamide and it even enhanced its anticancer properties [164]. Then, in clinical trial NCT02807805, niclosamide was used in an orally bioavailable niclosamide/PDMX1001 formulation. This reformulated PDMX1001/niclosamide was reported, after oral administration, to be able to reach therapeutic concentrations consistent with preclinical studies [165]. One of the recent studies reported another approach for the reformulation of niclosamide. A specific niclosamide formulation was prepared to be used not for oral administration, but for topical delivery to the upper skin layers—niclosamide-loaded liposomal thermogel for the treatment of melanoma [166].

Based on the Biopharmaceutical Classification System, solubility and intestinal permeability are referred to as the two most distinctive factors affecting the oral absorption of drugs [167]. Niclosamide was classified as a BCS Class II drug, which points to its low aqueous solubility and limited absorption which depends on dissolution rate [168]. In general, salicylanilides are weakly acidic, highly lipophilic compounds [17]. Low water solubility is typical not only for niclosamide [169], but also for other salicylanilides such as rafoxanide (whose solubility in water was reported to be less than 0.1 μg/mL) [170], closantel [171] and for nitrothiazolyl-salicylamide nitazoxanide as well [172]. As it was already mentioned, to overcome challenges connected to the undesirable physicochemical properties that influence oral bioavailability, a number of studies reported diverse innovative formulations of salicylanilides. A recent extensive review summarizes the wide range of studies devoted to the development of nanostructured delivery systems containing niclosamide as nanocrystals, polymeric or lipid nanoparticles, etc. [173]. However, there are already more recent studies reporting new formulations of niclosamide, as a development of amorphous solid dispersion [174] or reformulated niclosamide as a niclosamide stearate prodrug therapeutic [175]. Moreover, similar reformulation efforts already exist in the case of other salicylanilides such as closantel [171], rafoxanide [170] and even in the case of structurally related nitazoxanide [172,176].

## 8. Conclusions and Limitations

Drugs with a salicylanilide core structure have been studied for many years for their possible repositioning for the treatment of diverse types of tumors. The number of studies contributing to the explanation of their potential anticancer effect potential is still increasing. Recent findings indicate that the spectrum of influenced signaling pathways and other cellular targets is very wide. Other proposed mechanisms such as enhancement of anticancer immunity are still emerging. There are also hypotheses that the targeting specific signaling pathway is a secondary effect of mitochondrial uncoupling.

Nowadays, not only niclosamide but also other salicylanilides or structurally related compounds are intensively studied for their promising anticancer properties, and alongside niclosamide, nitazoxanide is already being included in cancer clinical trials. A major problem that still complicates their future use in cancer therapy is their generally inappropriate pharmacokinetic properties, mainly poor oral bioavailability. That is why there are already several studies searching for solutions to improve their pharmacokinetic properties.

In conclusion, we can state that salicylanilides represent a very interesting group of compounds with the potential of anticancer action, which deserve further attention.

## Figures and Tables

**Figure 1 ijms-24-01728-f001:**
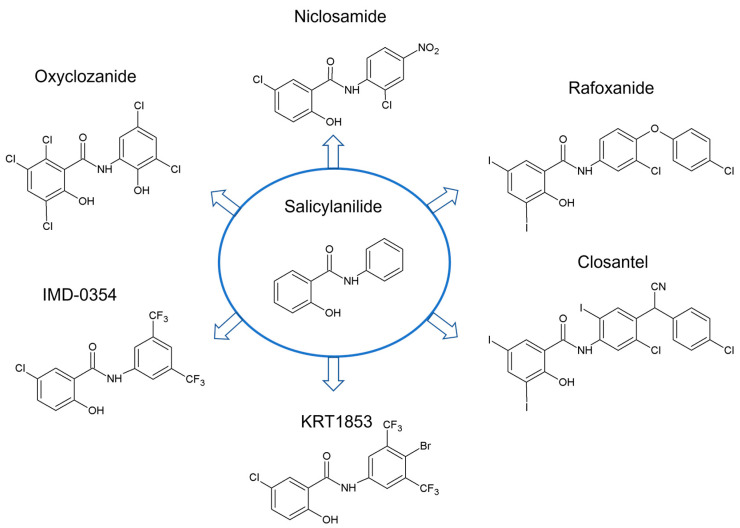
Structures of salicylanilides investigated for their potential anticancer properties.

**Figure 2 ijms-24-01728-f002:**
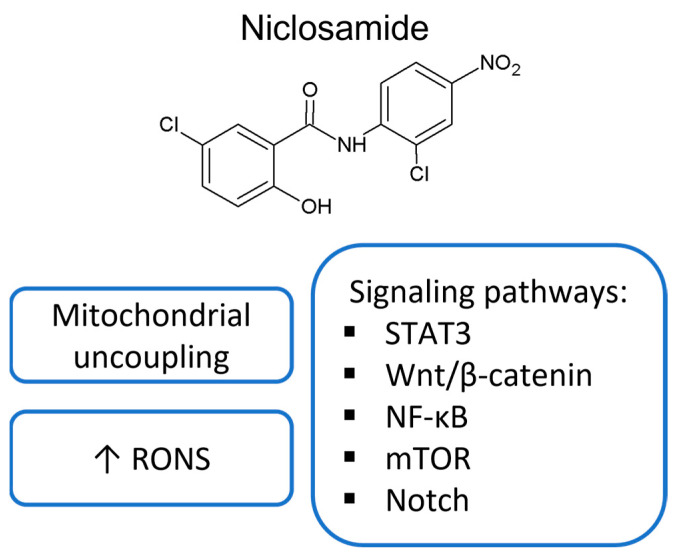
Summary of the most frequently proposed mechanisms of niclosamide’s anticancer effect. RONS, reactive oxide and nitrogen species.

**Figure 3 ijms-24-01728-f003:**
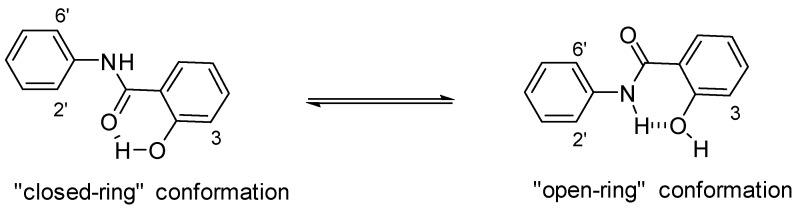
Conformational equilibrium in salicylanilide molecules. Two possible intramolecular hydrogen bond formations resulting in closed-ring or open-ring conformation [52].

**Figure 4 ijms-24-01728-f004:**
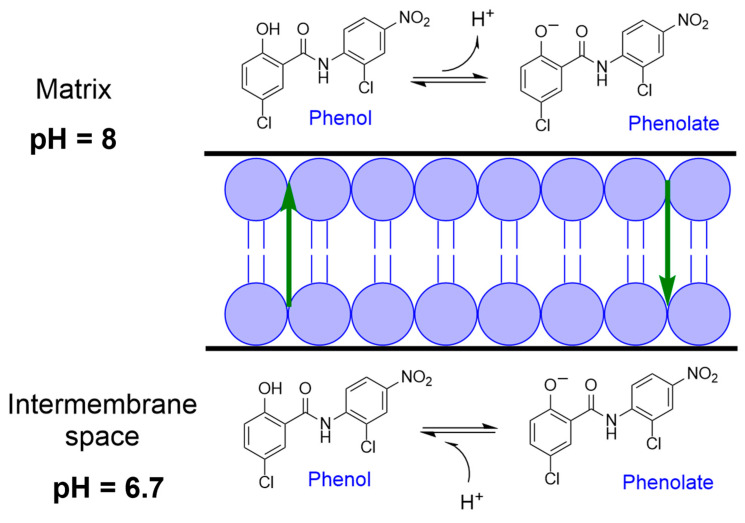
Protonophore activity of niclosamide in the inner mitochondrial membrane. Scheme was created according to [60,61].

**Figure 5 ijms-24-01728-f005:**
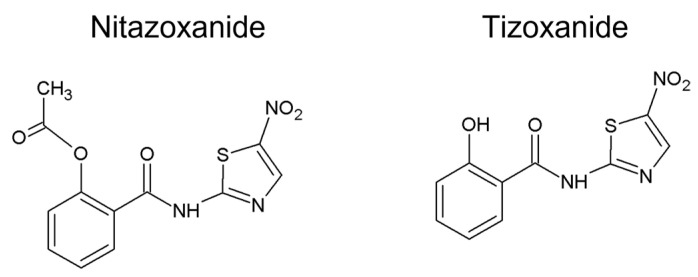
Structures of nitazoxanide and its metabolite tizoxanide.

**Figure 6 ijms-24-01728-f006:**
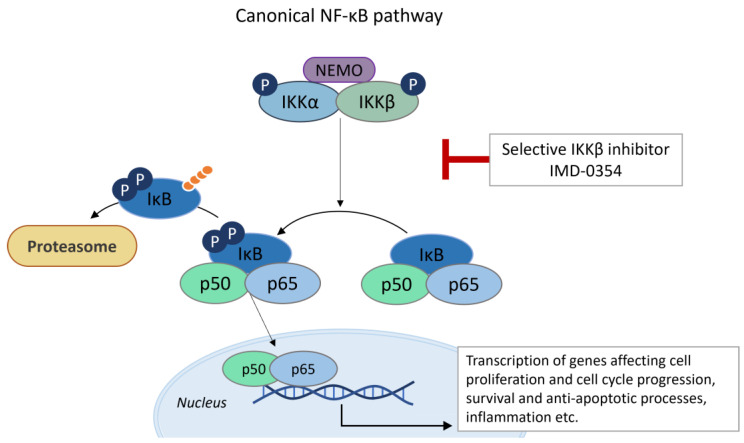
Canonical NF-κB pathway as possible target of selective inhibitor of IKKβ IMD-0354. Upon wide variety of activating stimuli, trimeric complex that consists of catalytic subunits, IKKα and IKKβ, and a regulatory subunit NEMO could phosphorylate two serine residues of IκB bound to NF-κB dimers. NF-κB occurs mainly in the form of p50/p65 dimers. This phosphorylation leads to ubiquitination of IκB and its proteasome degradation. Thus NF-κB is translocated into the nucleus and modulates gene transcription. Scheme was created according to [133,137,138].

**Figure 7 ijms-24-01728-f007:**
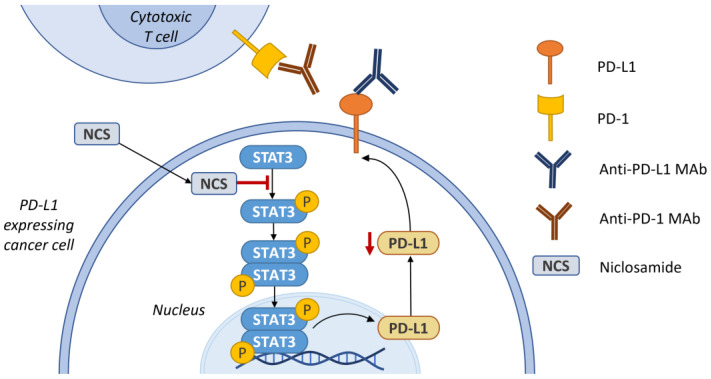
Synergistic activity of niclosamide with immune checkpoint inhibitors e.g., anti-PD-L1 or anti-PD-1 monoclonal antibodies. Niclosamide suppresses phosphorylation of STAT3 and thus downregulates the expression of PD-L1. Scheme was created according to [104,161].

**Table 1 ijms-24-01728-t001:** List of studies focused on anticancer properties, including proposed mechanism of action and detailed effects of salicylanilide compounds closantel, oxyclozanide, rafoxanide, IMD-0354, KRT1853 and nitrothiazolyl-salicylamide derivative nitazoxanide.

Salicylanilide	Proposed Mechanism of Action	Model/Methods	Effects	Reference
Closantel	Antiangiogenic activity	Ramos, Hela, PANC-1, and HepG2 cell xenografts in zebrafish	Inhibited growth of xenotransplanted cells in zebrafish	[31]
Closantel	B-Raf V600E inhibition	ELISA-based assay molecular docking with halogen bonding scoring function	IC_50_ = 1.90 μM	[32]
Closantel	Mitochondrial dysfunction	HCT116 and HT-29 cells grown in monolayer and in vitro 3D model; tumor spheroids of HCT116 GFP and HT-29 GFP cell lines	Inhibited colony formation, inhibited mitochondrial respiration, decreased oxygen consumption, induced depolarization of mitochondrial membrane	[6]
Closantel	Taspase1 inhibition	Cell-free system (cfs-Taspase1), *E. coli*-produced Taspase1, HEK293T cells expressing GFP-CS2-BFP and Taspase1	IC_50_ = 1.6 μM 1 (cfs-produced Taspase1) IC_50_ = 3.9 μM (*E. coli*-produced Taspase1)	[33]
Oxyclozanide	Mitochondrial uncoupling	MC38, HCT116 and C2C12 cell lines; MC38 xenografts in NSG mice	In vitro: decreased oxygen consumption, AMPK activation, inhibition of the mTOR activity In vivo: diminished hepatic metastases	[4]
Rafoxanide	CDK4/6 dual-inhibition	A375 and A431 cell lines; A375 xenografts in BALB/C nude mice	In vitro: decreased expression of CDK4/6, cyclin D, Rb, pho-CDK4/6, pho-Rb induced apoptosis, cell cycle arrest in G1 phase In vivo: reduced the growth of tumors	[34]
Rafoxanide	DNA damage responses, suppression of p38 MAPK pathway	Multiple myeloma cell lines; H929 xenografts in BALB/C nude mice	In vitro: antiproliferative effect, induced apoptosis, decreased mitochondrial membrane potential MMP, cell cycle arrest in G1 phase; synergistic activity with bortezomib or lenalidomide In vivo: tumor growth inhibition	[7]
Rafoxanide	PI3K /Akt/mTOR pathway suppression	SGC-7901 and BGC-823 cell lines; SGC-7901 xenografts in BALB/c nude mice	In vitro: antiproliferative effect, cell cycle arrest in G1 phase, promoted apoptosis and autophagy In vivo: tumor growth inhibition	[35]
Rafoxanide	B-Raf V600E inhibition	ELISA-based assay molecular docking with halogen bonding scoring function	IC_50_ = 0.07 μM	[32]
Rafoxanide	Endoplasmic reticulum stress induction	HCT-116 and HT-29 cell lines, human CRC Explants	In vitro: decreased p-ERK expression, antiproliferative effect, cell cycle arrest in G1 phase, induced cell death In vivo: reduced colonic tumorigenesis, induced apoptosis, eIF2α phosphorylation	[36]
Rafoxanide	*bona fide* immunogenic cell death (ICD) induction	HCT-116, DLD1 and CT26 cell lines, tumor vaccination model rafoxanide-treated CT26 cells	In vitro: induction of ecto-calreticulin exposure, adenosine triphosphate (ATP)/high mobility group box 1 (HMGB1) release In vivo: no visible sign of tumor growth or only tiny tumor masses in animals vaccinated with rafoxanide-treated CT26 cells	[37]
Rafoxanide	Restoration of sensitivity to TRAIL	DLD-1 and SW480 cell lines, BALB/c mice with CT26-derived grafts	In vitro: decreased c-FLIP and surviving expression In vivo: synergism with TRAIL in inhibiting the growth of CT26-derived tumors	[38]
Rafoxanide	PTEN/PI3K/AKT and JNK/c-Jun pathways suppression	Diffuse large B-cell lymphoma (DLBCL) cell lines (NU-DUL-1 and OCI-LY8 cells), nude mice with OCI-LY8-derived grafts	In vitro: induced apoptosis, decreased MMP, increased reactive oxygen species (ROS) generation, induced DNA damage In vivo: inhibition of tumor growth, induction of TUNEL positive cells production	[39]
IMD-0354	NF-κB inhibition	HMC-1 cell line	decreased translocation of NF-κB to the nucleus, antiproliferative effect, cell cycle arrest in G1 phase, decreased cyclin D3 expression and pRb phosphorylation	[40]
IMD-0354	NF-κB inhibition	MCF-7, MDA-MB-231, HMC1-8 cell lines, MDA-MB-231 xenografts in BALB/c nude mice	In vitro: decreased NF-κB activity without affecting Akt phosphorylation, antiproliferative effect, cell cycle arrest, down-regulation of D-type cyclins, decreased Rb phosphorylation and expression Bcl family proteins and MDM2 In vivo: suppressed tumor growth without any serious side effects and toxicity	[8]
IMD-0354	NF-κB inhibition	Chronic lymphocytic leukemia (CLL) cell samples from untreated CLL patients	induced apoptosis, decreased survival index in vitro, inhibition of NF-κB activation, induction of apoptotic-related gene expression	[41]
IMD-0354	NF-κB inhibition	Peripheral blood mononuclear cells (PBMCs) from adult T-cell leukemia (ATL) patients, ATL-derived cell lines, human T-cell leukemia virus type I (HTLV-I)-infected, and HTLV-I free T-cell lines, ATL-43Tb(-) xenografts NOG mice	In vitro: selective decrease of viability of CD4+CD25+ primary ATL cells in vitro (IC50 = 2.87 µM; caspase 3/7 activation), decreased growth and transcriptional activity of NF-κB in HTLV-I-infected cells In vivo: Suppressed tumor growth	[42]
IMD-0354	IKKβ inhibition	MSTO-211H, NCI-H2052 and NCI-H28 cell lines, MSTO-211H xenografts in BALB/c-nu/nu mice or SCID mice	In vitro: decreased activation of NF-κB by IKKβ inhibition, cell cycle arrest in G1 phase, decreased cyclin D3 expression In vivo: Suppressed tumor growth	[43]
IMD-0354	IKKβ inhibition	Panc-1, PK8 cell lines, Panc-1 xenografts in NOG mice	In vitro: decreased NF-κB activity, antiproliferative effect In vivo: suppressed tumor growth without toxicity	[44]
IMD-0354	Modulation of glutamine metabolism, mTOR signaling suppression	A431 and A375 cell lines; prostate, pancreatic, colon, lung, and kidney tumor–derived cell lines; SW1 melanoma xenografts in C3H mice	In vitro: inhibition of SLC1A5 localization at the plasma membrane, decreased glutamine uptake, decreased glutamine-dependent amino acid pathways, decreased mTOR signaling, antiproliferative effect, induced apoptosis and autophagy; dose-dependent viability inhibition In vivo: xenograft growth inhibition, decreased mTOR activity, induced apoptosis	[45]
IMD-0354 KRT1853	TMPRSS4 serine protease inhibition	Prostate, colon, and lung cancer cell lines	In vitro: decreased cancer cell invasion, migration, and proliferation in TMPRSS4-expressing cells, induced apoptosis, decreased TMPRSS4-mediated signaling activity	[9]
KRT1853	TMPRSS4 serine protease inhibition	DU145 and HCT116 cell xenografts in nude mice	In vivo: suppressed tumor growth	[9]
Nitazoxanide	Mitochondrial dysfunction AMPK pathway activation, c-Myc, mTOR, and Wnt signaling downregulation	HCT116 and HT-29 cells grown in monolayer and in vitro 3D model –tumor spheroids of HCT116 GFP and HT-29 GFP cell line HCT116 xenografts in NMRI nu/nu mice	In vitro: inhibited colony formation, inhibited mitochondrial respiration, decreased oxygen consumption, induced depolarization of mitochondrial membrane, AMPK, downregulation In vivo: potentiation of irinotecan effect inducing reduction in tumor growth and volume	[6]
Nitazoxanide	Inhibition of Wnt/β-catenin signaling through PAD2, independent of GSK3β and APC	293Ft and SW480 cell line Murine model for familial adenomatous polyposis (*Apc^min/+^* mice) with spontaneously generate tumors	In vitro: dose-dependent decrease of β-catenin, stabilization of PAD2 and increase in protein citrullination In vivo: Decreased numbers of micro- and macroadenomas, reduction of β-catenin	[46]
Nitazoxanide	c-Myc inhibition	Quantitative high throughput screening for c-Myc inhibition, SKBR3 cell line, SKBR3 cell xenografts in nude mice (nu/nu)	80.5% c-Myc activity inhibition at 10 µM, inhibition of c-Myc in SKBR3 cells (IC_50_ ~100 nM) (luciferase activity assay), inhibition of phospho c-Myc level cell lines SKBR3 (IC_50_ = 122 nM), lymphoma (IC_50_ = 440 nM) and osteosarcoma-derived cell line In vivo: decreased c-Myc activity and suppressed tumor growth	[47]
Nitazoxanide	ING1 upregulation (increased expression and decreased cleavage)	LN229, U87, A172, and HUVEC cell lines, LN229 xenografts in BALB/C nude mice (s.c. injected), orthotopic brain tumor model of LN229 xenografts in BALB/C nude mice (injected into *caudate nucleus*)	In vitro: Antiproliferative effect, cycle arrest in G1 phase, blockage of late-stage autophagic flux In vivo: glioma growth inhibition, increased levels of ING1, LC3 and SQSTM1; inhibition of intracranial tumor growth, survival time prolongation	[10]
Nitazoxanide	Downregulation of Wnt/β-catenin/GSK-3β signaling	HCT-116 cell line, (FHC normal colon cell line), 1,2-dimethylhydrazine (DMH) induced colon cancer in Swiss albino mice	In vitro: Cytotoxic activity (HCT-116 IC_50_ = 11.07 μM; FHC IC_50_ = 48.4 μM), induced apoptosis, increased mRNA expression of p53, BAX, caspases 3, 8 and 9 In vivo: decreased Wnt, β-catenin and GSK-3β levels, decrease in number of PCNA positive nuclei, reduction of pathologic signs based on histopathologic scoring data	[48]
Nitazoxanide	β-catenin inhibition, synergistic effects with obeticholic acid (OCA)	SW403, SW480, DLD-1, and HT-29 cell lines, SW403 and SW480 xenografts in BALB/c-nude mice	In vitro: growth inhibition IC_50_ = 2.764 μM (SW403), IC_50_ = 2.294 μM (SW480), IC_50_ = 2.149 μM (DLD-1), IC_50_ = 1.930 μM (HT-29); decreased β-catenin expression, synergistic effects with OCA (repressed colony formation, cell cycle arrest in G1 phase, induced apoptosis) In vivo: Synergistic effect with OCA inducing tumor growth reduction	[49]

**Table 2 ijms-24-01728-t002:** List of cancer clinical trials with niclosamide and nitazoxanide.

Name of Drug	Type of Cancer	Phase	Clinical Trials.gov Identifier	Current Clinical Trial Status	Location	Reference
Niclosamide	Colon Cancer	Phase 1	NCT02687009	Terminated (low accrual)	Duke University, NC, US	-
Niclosamide	Colorectal Cancer	Phase 2	NCT02519582	Unknown †	Charite University, Berlin, Germany	[14]
Niclosamide (w/enzalutamide)	Castration-Resistant Prostate Cancer	Phase 1	NCT03123978	Active, not recruiting	University of California Davis, Comprehensive Cancer Center, CA, US	-
Niclosamide (w/abiraterone acetate and prednisone)	Hormone-Resistant Prostate Cancer	Phase 2	NCT02807805	Active, not recruiting	University of California Davis, Comprehensive Cancer Center, CA, US	-
Niclosamide (w/ enzalutamide)	Castration-Resistant Prostate Carcinoma Metastatic Prostate Carcinoma Recurrent Prostate Carcinoma	Phase 1	NCT02532114	Completed	Fred Hutch/University of Washington Cancer Consortium, WA, US	[13]
Niclosamide	Familial Adenomatous Polyposis	Phase 2	NCT04296851	Recruiting	Department of Internal Medicine, Yonsei University College of Medicine Seoul, Republic of Korea	-
Niclosamide	Acute Myeloid Leukemia	Phase 1	NCT05188170	Recruiting	Stanford University, CA, US	-
Nitazoxanide (Antibacterial Agents Antifungal Agents Antiprotozoal Agents)	Neoplasms	Phase 2	NCT02366884	Recruiting	Dr. Frank Arguello Cancer Clinic, Instituto de Ciencia y Medicina Genomica, Mexico	-

† Study has passed its completion date and status has not been verified in more than two years.

## Data Availability

Not applicable.

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
