# Peer review of "Salicylanilides and Their Anticancer Properties"

_ijms, 2023, doi:10.3390/ijms24021728_

Round 1

Reviewer 1 Report

Reviewer’s Comments:

The manuscript “Salicylanilides and Their Anticancer Properties” is a very interesting work. In this work, alicylanilides are pharmacologically active compounds with a wide spectrum of biological effects. Halogenated salicylanilides, which have been used for decades in human and veterinary medicine as anthelmintics, have recently emerged as candidates for drug repurposing in oncology. The most prominent example of salicylanilide anthelmintic, which is intensively studied for its potential anticancer properties, is niclosamide. Nevertheless, recent studies have discovered the extensive anticancer potential in several other salicylanilides. This potential of their anticancer action is mediated most likely by diverse mechanisms of action such as uncoupling of oxidative phosphorylation, inhibition of protein tyrosine kinase epidermal growth factor receptor, modulation of different signaling pathways such as Wnt/β-catenin, mTORC1, STAT3, NF-κB and Notch signaling pathways or induction of B-Raf V600E inhibition. While I believe this topic is of great interest to our readers, I think it needs major revision before it is ready for publication. So, I recommend this manuscript for publication with major revisions.

1. In this manuscript, the authors did not explain the importance of the Anticancer Properties in the introduction part. The authors should explain the importance of Anticancer Properties.

2) Title: The title of the manuscript is not impressive. It should be modified or rewritten it.

3) Correct the following statement “Here we provide a comprehensive overview of the current knowledge about the proposed mechanisms of action of anticancer activity of salicylanilides based on preclinical in vitro and in vivo studies. We summarize proposed mechanisms of anticancer action or structural requirements for such an activity of niclosamide and other salicylanilides”.

4) Keywords: The Anticancer Properties is missing in the keywords. So, modify the keywords.

5) Introduction part is not impressive. The references cited are very old. So, Improve it with some latest literature like 10.1016/j.molstruc.2021.131145, 10.3389/fchem.2022.1023316

6) The authors should explain the following statement with recent references, “During mitochondrial oxidative phosphorylation (OXPHOS), a nutrient oxidation is coupled to ATP production through reactions in the electron transport chain (ETC)”.

7) Add space between magnitude and unit. For example, in synthesis “21.96g” should be 21.96 g. Make the corrections throughout the manuscript regarding values and units.

8) The author should provide reason about this statement “In a number of different types of solid tumors, EGFR deregulation is observed, which leads to uncontrolled activation of the receptor and associated signaling cascades, therefore they become an attractive target for anticancer therapy”.

9. Comparison of the present results with other similar findings in the literature should be discussed in more detail. This is necessary in order to place this work together with other work in the field and to give more credibility to the present results.

10) Conclusion part is very long. Make it brief and improve by adding the results of your studies.

11) There are many grammatic mistakes. Improve the English grammar of the manuscript.

Author Response

We thank the reviewer for recognizing the potential of our work and for the valuable suggestions on how to improve the manuscript. We have reworked the manuscript and included new information that respond to the critique raised by the reviewer.

Below, our responses to the reviewer, highlighted by using bold italics text, are available.

Reviewer 1

Comments and Suggestions for Authors:

The manuscript “Salicylanilides and Their Anticancer Properties” is a very interesting work. In this work, salicylanilides are pharmacologically active compounds with a wide spectrum of biological effects. Halogenated salicylanilides, which have been used for decades in human and veterinary medicine as anthelmintics, have recently emerged as candidates for drug repurposing in oncology. The most prominent example of salicylanilide anthelmintic, which is intensively studied for its potential anticancer properties, is niclosamide. Nevertheless, recent studies have discovered the extensive anticancer potential in several other salicylanilides. This potential of their anticancer action is mediated most likely by diverse mechanisms of action such as uncoupling of oxidative phosphorylation, inhibition of protein tyrosine kinase epidermal growth factor receptor, modulation of different signaling pathways such as Wnt/β-catenin, mTORC1, STAT3, NF-κB and Notch signaling pathways or induction of B-Raf V600E inhibition. While I believe this topic is of great interest to our readers, I think it needs major revision before it is ready for publication. So, I recommend this manuscript for publication with major revisions.

1) In this manuscript, the authors did not explain the importance of the Anticancer Properties in the introduction part. The authors should explain the importance of Anticancer Properties.

We have edited Introduction part of the manuscript based on the recommendations from Reviewer 1 and 3 by adding additional literature references focused on the topic of anticancer properties of salicylanilides. Thus, we have strengthened the explanation of their potential anticancer effects. Introduction part of the manuscript also deals with the importance of finding novel anticancer drugs that specifically interfere with processes typical of tumor cells. Moreover, the importance of the drug repurposing process is also emphasized, which can bring a number of benefits compared to the standard process of developing a new anticancer drug.

2) Title: The title of the manuscript is not impressive. It should be modified or rewritten it.

Here we propose possible alternative titles for our manuscript to the current title “Salicylanilides and Their Anticancer Properties”:

“Can we consider salicylanilides as potentially useful in cancer therapy?” or “Can salicylanilides be seen as interesting for cancer therapy?” or “Salicylanilides - molecules with an antitumor potential?”

We would like to leave this decision to the editors of our article.

3) Correct the following statement “Here we provide a comprehensive overview of the current knowledge about the proposed mechanisms of action of anticancer activity of salicylanilides based on preclinical in vitro and in vivo studies. We summarize proposed mechanisms of anticancer action or structural requirements for such an activity of niclosamide and other salicylanilides”.

We have corrected that statement according to Reviewer's suggestion, and the changes to that statement are highlighted using yellow marked text.

4) Keywords: The Anticancer Properties is missing in the keywords. So, modify the keywords.

We have added this phrase among the keywords; it is highlighted in yellow.

5) Introduction part is not impressive. The references cited are very old. So, Improve it with some latest literature like 10.1016/j.molstruc.2021.131145, 10.3389/fchem.2022.1023316

As both, the Reviewer 1 and 3 recommended us to provide additional publications focused on this topic, we have included more citations of recent publications into the introduction part. Nevertheless, we would like to preserve some of the older references in the manuscript as they were included into the text on purpose, because they refer to unique studies that either documented the discovery of a specific effect of salicylanilides and their conclusions are still valid today, or they document the clinical use of the studied substances, that describes the chronology of the development of knowledge about the issue.

However, we decided not to include the recommended publications, as none fits the topic addressed in our manuscript perfectly.

6) The authors should explain the following statement with recent references, “During mitochondrial oxidative phosphorylation (OXPHOS), a nutrient oxidation is coupled to ATP production through reactions in the electron transport chain (ETC)”.

We have reformulated this statement using additional references and explained it in more detail (it is highlighted in yellow in the manuscript).

7) Add space between magnitude and unit. For example, in synthesis “21.96g” should be 21.96 g. Make the corrections throughout the manuscript regarding values and units.

Although we tried to find the part of the text mentioned by the Reviewer in our manuscript (specifically "synthesis 21.96g"), we were unable to find it and therefore we assume that our paper does not contain it. Nevertheless, based on this comment, we adjusted the missing spaces in our text.

8) The author should provide reason about this statement “In a number of different types of solid tumors, EGFR deregulation is observed, which leads to uncontrolled activation of the receptor and associated signaling cascades, therefore they become an attractive target for anticancer therapy”.

We have reformulated this statement and explained it in more detail (it is highlighted in yellow) and added more references as well. The reason why we included this statement in the text, is that dysregulation of EGFR (e.g., increased expression of the HER2 receptor in a certain type of breast cancer) is typical for some types of tumors. Inhibiting the EGFR activation has been proven to be effective in the therapy of such tumors. Some salicylanilides have been reported to inhibit TK EGFR; therefore, we have included this statement in the text.

9) Comparison of the present results with other similar findings in the literature should be discussed in more detail. This is necessary in order to place this work together with other work in the field and to give more credibility to the present results.

We are not sure this remark was intended for our work, as we do not present newly obtained results in the manuscript but summarize the already known properties of salicylanilides in terms of their anticancer activity. This information is discussed in the context of current knowledge within the entire presented text.

10) Conclusion part is very long. Make it brief and improve by adding the results of your studies.

Thank you for your worthwhile suggestion. We made the conclusion more concise. We mentioned the results of our studies related to the revised topic in the text of manuscript, nevertheless the prevailing number of studies devoted to individual aspects of the problem related to salicylanilides and their antitumor properties are from other authors. In the conclusion, the most important aspects of the results of the individual studies are briefly mentioned, which are discussed in more detail in the individual chapters throughout the manuscript.

11) There are many grammatic mistakes. Improve the English grammar of the manuscript.

We used the services of a native speaker and had the entire manuscript corrected. Thus, we have made few grammar or stylistic improvements in the current version.

Reviewer 2 Report

The article "Salicylanilides and Their Anticancer Properties" is a good piece of article. I really enjoyed to read this article. Well overall the article is well written and merit publication, But i have some minor suggestions i hope considering these suggestions will further enhance the quality of the article. As this article is submitted to IJMS which worth the article encompassing mechanisms.

1. It would be great if the authors can draw a detailed mechanisms diagrams for signaling pathways like STAT3, NfKB , mTORetc. As the mechanisms diagrams are the backbone of the article. It really attracts the eyes of the readers. 

2.For table 2 i would suggest authors should add more clear information about clinical trails status like if it is Phase 1, then it is still going on or it is successfully completed/terminated/failed/ or recruiting or what, and also add one more column mentioning who is doing these clinical trails (Country, pharmaceuticals or any University)

3. The second last paragraph of the conclusion contains the limitations of this study which is really a good aspect so the authors should mention the title conclusions and limitations.

Author Response

We thank the reviewer for recognizing the potential of our work and for the valuable suggestions on how to improve the manuscript. We have reworked the manuscript and included new information that respond to the critique raised by the reviewer.

Below, our responses to the reviewer, highlighted by using bold italics text, are available.

Reviewer 2

Comments and Suggestions for Authors:

The article "Salicylanilides and Their Anticancer Properties" is a good piece of article. I really enjoyed to read this article. Well overall the article is well written and merit publication, But i have some minor suggestions i hope considering these suggestions will further enhance the quality of the article. As this article is submitted to IJMS which worth the article encompassing mechanisms.

1) It would be great if the authors can draw a detailed mechanisms diagrams for signaling pathways like STAT3, NfKB , mTORetc. As the mechanisms diagrams are the backbone of the article. It really attracts the eyes of the readers. 

As recommended by both, reviewers 2 and 3, we decided to create and supplement two figures (Figure 6, Figure 7) that illustrate the signaling pathways, inhibited by salicylanilides.

2) For table 2 i would suggest authors should add more clear information about clinical trails status like if it is Phase 1, then it is still going on or it is successfully completed/terminated/failed/ or recruiting or what, and also add one more column mentioning who is doing these clinical trails (Country, pharmaceuticals or any University)

Thank you for your valuable suggestion. We broaden the information in Table 2. Such a change is highlighted in blue in the text of the manuscript.

3) The second last paragraph of the conclusion contains the limitations of this study which is really a good aspect so the authors should mention the title conclusions and limitations.

We are glad that the Reviewer appreciated the information about the limitations presented in the Conclusions. Suppose the IJMS editors allow us to change the fixed title of this chapter (compared to the recommended article template). In that case, we can rename the last chapter of the text "Conclusions and limitations". We leave this decision to the editors of our article.

NOTE January 12: After discussing point #3 with the editor, we changed the title of the last chapter of the manuscript to “Conclusions and Limitations”.

Reviewer 3 Report

In the present review article titled “ Salicylanilides and Their Anticancer Properties”, authors vigorously reviewed the anticancer potential of Salicylanilides. Authors discussed the structural features of Salicylanilides related to their biological activity and how these Salicylanilides modulates different cellular pathways. There are few points need to be revisit.

1.      Line 40, citation is missing.

2.      Authors can generate a figure involving different signaling pathways and Salicylanilides where they act.

3.      Some of article are missing in this present article. For example: https://pubmed.ncbi.nlm.nih.gov/31530650/; https://www.ncbi.nlm.nih.gov/pmc/articles/PMC6109047/; https://pubmed.ncbi.nlm.nih.gov/26469835/. Please revisit the literature.

4.      As authors discussed the “structural features of salicylanilides related to their biological activity”. Author can analyse the physicochemical properties (polarity, solubility, saturation, flexibility, lipophilicity and size) of different discussed salicylanilides and highlights the points where researches can modify it for better anticancer activity.

Author Response

We thank the reviewer for recognizing the potential of our work and for the valuable suggestions on how to improve the manuscript. We have reworked the manuscript and included new information that respond to the critique raised by the reviewer.

Below, our responses to the reviewer, highlighted by using bold italics text, are available.

Reviewer 3

Comments and Suggestions for Authors:

In the present review article titled “Salicylanilides and Their Anticancer Properties”, authors vigorously reviewed the anticancer potential of Salicylanilides. Authors discussed the structural features of Salicylanilides related to their biological activity and how these Salicylanilides modulates different cellular pathways. There are few points need to be revisit.

1) Line 40, citation is missing.

Thank you for the notice. We have added the missing citations.

2) Authors can generate a figure involving different signaling pathways and Salicylanilides where they act.

As recommended by both, reviewers 2 and 3, we decided to create and supplement two figures (Figure 6, Figure 7) that illustrate the signaling pathways, inhibited by salicylanilides.

3) Some of article are missing in this present article. For example: https://pubmed.ncbi.nlm.nih.gov/31530650/;

https://www.ncbi.nlm.nih.gov/pmc/articles/PMC6109047/; https://pubmed.ncbi.nlm.nih.gov/26469835/. Please revisit the literature.

Thank you very much for the valuable suggestions. We have further expanded the list of references by articles the Reviewer proposed. To broaden the list of publications, we have also added references to papers describing the anticancer potential of salicylanilides e.g., those regarding the inhibitory activity of niclosamide against the Wnt signaling pathway or the potential of closantel to inhibit Taspase1; highlighted in green (Table 1).

4) As authors discussed the “structural features of salicylanilides related to their biological activity”. Author can analyse the physicochemical properties (polarity, solubility, saturation, flexibility, lipophilicity and size) of different discussed salicylanilides and highlights the points where researches can modify it for better anticancer activity.

Based on this suggestion, we added a paragraph regarding important physicochemical properties that may influence the future clinical use of salicylanilides in the anticancer indication. We also present approaches to improve these properties and the possibility of reformulating salicylanilides, most often, to improve their bioavailability. Such a paragraph is highlighted in green in the final text.
